

# A post-processed carbon flux dataset for 34 eddy covariance flux sites across the Heihe River Basin, China

Xufeng Wang[1], Tao Che[1]*, Jingfeng Xiao[2], Tonghong Wang[1,3], Junlei Tan[1], Yang Zhang[1], Zhiguo Ren[1], Liying Geng[1], Haibo Wang[1], Ziwei Xu[4], Shaomin Liu[4], Xin Li[5]

[1] Key Laboratory of Remote Sensing of Gansu Province/Heihe Remote Sensing Experimental Research Station/Key Laboratory of Cryospheric Science and Frozen Soil Engineering, Northwest Institute of Eco-Environment and Resources, Chinese Academy of Sciences, Lanzhou 730000, China.

[2] Earth Systems Research Center, Institute for the Study of Earth, Oceans, and Space, University of New Hampshire, Durham, NH 03824, USA.

[3] School of Geography and Environmental Sciences, Northwest Normal University, Lanzhou 730000, China.

[4] State Key Laboratory of Remote Sensing Science, School of Geography, Beijing Normal University, Beijing 100101, China.

[5] Institute of Tibetan Plateau Research, Chinese Academy of Sciences, Beijing 100101, China.

*Correspondence to*: Tao Che (chetao@lzb.ac.cn)

**Abstract.** The eddy covariance (EC) technique is currently the most widely used method for measuring carbon exchange between terrestrial ecosystems and the atmosphere at the ecosystem scale. Using this technique, a regional carbon flux network comprising a total of 34 sites has been established in the Heihe River Basin (HRB) in Northwest China. This network has been

measuring the net ecosystem exchange (NEE) of $CO_2$ for a variety of vegetation types. In this study, we compiled and post-processed half-hourly flux data from these 34 EC flux sites in the HRB to create a continuous, homogenized time series dataset. We employed standardized processing procedures to fill data gaps in meteorological and NEE measurements at half-hourly intervals. NEE measurements were also partitioned into gross primary production (GPP) and ecosystem respiration (Reco). Furthermore, half-hourly meteorological and NEE data were aggregated to daily, weekly, monthly, and yearly timescales. As

a result, we produced a continuous carbon flux and auxiliary meteorological dataset, which includes 18 sites with continuous multi-year observations and 16 sites observed only during the 2012 growing season, amounting to a total of 1,513 site-months. Using the post-processed dataset, we explored the temporal and spatial characteristics of carbon exchange in the HRB. In the diurnal variation curve, GPP, NEE, and Reco peak later for ecosystems in the artificial oasis (cropland and wetland) compared to those outside the artificial oasis (grassland, forest, woodland, and Gobi/desert). Seasonal NEE, GPP, and Reco peak in early

July for grassland, forest, woodland, and cropland but remain close to zero throughout the year for gobi/desert. In the last decade, NEE of wetlands significantly increased, while NEE for other ecosystems did not exhibit significant trends. Annual NEE, GPP, and Reco are significantly higher for sites inside the artificial/natural oasis compared to those outside the oasis. This post-processed carbon flux dataset has numerous applications, including exploring the carbon exchange characteristics of alpine and arid ecosystems, analyzing ecosystem responses to climate extremes, conducting cross-site synthesis from

regional to global scales, supporting regional and global upscaling studies, interpreting and calibrating remote sensing products, and evaluating and calibrating carbon cycle models.



## 1 Introduction

Terrestrial ecosystems absorb around 30% of anthropogenic carbon emissions (Friedlingstein et al., 2023) and thereby play a crucial role in the global carbon cycle. However, due to the complexity of the terrestrial ecosystems, efforts to quantify their carbon uptake capacity still face significant challenges. The eddy covariance (EC) technique is currently the most widely used method to measure the carbon exchange between terrestrial ecosystems and the atmosphere at the ecosystem scale (Baldocchi et al., 2001), providing insights into terrestrial carbon uptake capacity. Numerous regional and global carbon flux networks, such as FLUXNET, AmeriFlux, ICOS, AsiaFlux, TERN-OzFlux and ChinaFLUX, have been established to coordinate EC flux measurements across diverse terrestrial ecosystems. Despite the presence of over a thousand EC sites worldwide, these sites are predominantly located in North America, Europe, and East Asia (Pastorello et al., 2020). Many regions, like Northwest China and Central Asia, remain underrepresented, which hinders accurate quantification of carbon sinks in these areas and global-scale synthesis and upscaling studies.

The Heihe River Basin (HRB) is the second largest inland river basin in China and serves as an ideal experimental region for studying the carbon cycle in Northwest China (Cheng et al., 2014). Due to the significant gradients in elevation (ranging from 1,500 m a.s.l. downstream to 5,000 m a.s.l. upstream) and precipitation (from 50 mm downstream to 600 mm upstream), the HRB encompasses diverse landscapes, including snow/glacier, permafrost, alpine grassland, subalpine forest, irrigated cropland, riparian ecosystems, wetlands, and gobi/desert from the upstream to the downstream (Li et al., 2013). A carbon flux observation network in the HRB was established through two comprehensive field experiments: the Watershed Allied Telemetry Experimental Research (WATER) conducted from 2007 to 2010 (Li et al., 2009) and the Heihe Watershed Allied Telemetry Experimental Research (HiWATER) conducted from 2012 to 2017 (Li et al., 2013).

There are a total of 34 sites in the Heihe carbon flux network (Liu et al., 2018), among which 10 are long-term observation sites, while the rest are temporary sites that have been dismantled. The network started observing carbon flux data since 2008, and the quality controlled 30 min data is released annually on the National Tibetan Plateau Data Center. However, the released data contains numerous gaps due to instrument malfunctions and routine maintenance. Additionally, the net ecosystem exchange (NEE) has not been partitioned into gross primary productivity (GPP) and ecosystem respiration (Reco), two widely used carbon flux components in carbon cycle studies. These issues hinder the effective use of the dataset. To provide uniform and continuous carbon flux and auxiliary data, it is necessary to compile and post-process all the flux data in the HRB. Therefore, the objectives of this work are: 1) to effectively fill the gaps in carbon flux data and auxiliary meteorological data of the Heihe carbon flux network, and produce a high-quality, uniform, and continuous carbon flux dataset in the HRB; 2) to partition the half-hourly NEE measurements into GPP and Reco; and 3) to explore the diurnal, seasonal and inter-annual variability of carbon flux across diverse ecosystems in the HRB based on the gap-filled, partitioned dataset.

## 2 Carbon flux network in the HRB

The Heihe carbon flux network encompasses the main ecosystem types in the HRB, including alpine grassland, subalpine forest, wetland, irrigated cropland, riparian woody land, and gobi/desert (Fig. 1). Detailed information on these sites is provided in Table 1 and Table 2. The development of the Heihe carbon flux network has experienced three stages. The first stage spans from 2007 to 2011, during which the WATER experiment was conducted. During this period, the network comprised three sites: Arou, Guantan, and Yingke (Table 1) (Li et al., 2009). These three sites were dismantled in 2012.

The second stage spans from 2012 to 2015 when the HiWATER experiment was conducted (Li et al., 2013; Liu et al., 2018). During this period, the Heihe carbon flux network underwent comprehensive updates. In 2012, five new flux sites were established in and around the artificial oasis in the middle reaches of the HRB: Daman super site, Zhangye wetland site, Huazhaizi site, Bajitan site, and Shenshawo site (Table 1). Additionally, from May to September 2012, a flux matrix consisting of 17 sites was set up in the middle-stream artificial oasis (Table 2). In the summer of 2013, three flux sites were established in the upstream areas of the HRB: Arou super site, Dadongshu site, and Dashalong site (Table 1). Simultaneously, five sites



were set up in the downstream areas of the HRB: Sidaoqiao super site, Hunhelin site, Huyang site, Nongtian site, and Luodi site (Table 1).

The third stage spans from 2016 to the present, and the network was optimized to enhance its representativeness at the basin scale and to make maintenance more manageable (Liu et al., 2018). In 2016, four sites (Bajitan site, Shenshawo site, Luodi site, and Nongtian site) were dismantled, and two new flux sites were established: Huangmo site and Jingyangling site. Currently, ten sites are operational as long-term observing sites, with four in the upper reaches (Arou super site, Dashalong site, Dadongshu site, and Jingyangling site), three in the middle reaches (Daman super site, Zhangye wetland site, and Huazhaizi site), and three in the lower reaches (Sidaoqiao super site, Hunhelin site, and Huangmo site).

At each site of the carbon flux network in the HRB, flux and auxiliary meteorological factors, including $CO_2$ flux (Fc), latent heat flux (LE), sensible heat flux (H), downward solar radiation (Rg), air temperature (Ta), soil temperature (Ts), relative humidity (RH), soil water content (SWC), precipitation (P), and atmospheric pressure (PA), are recorded half-hourly or processed to half-hourly data. The 30 min data has been performed quality control (Xu et al., 2020; Liu et al., 2018; Che et al., 2019; Liu et al., 2023) and released at the TPDC (https://data.tpdc.ac.cn/zh-hans/topic/heihe). This released half-hourly carbon flux and auxiliary data include a lot of gaps, and the NEE data are not partitioned into GPP and Reco.

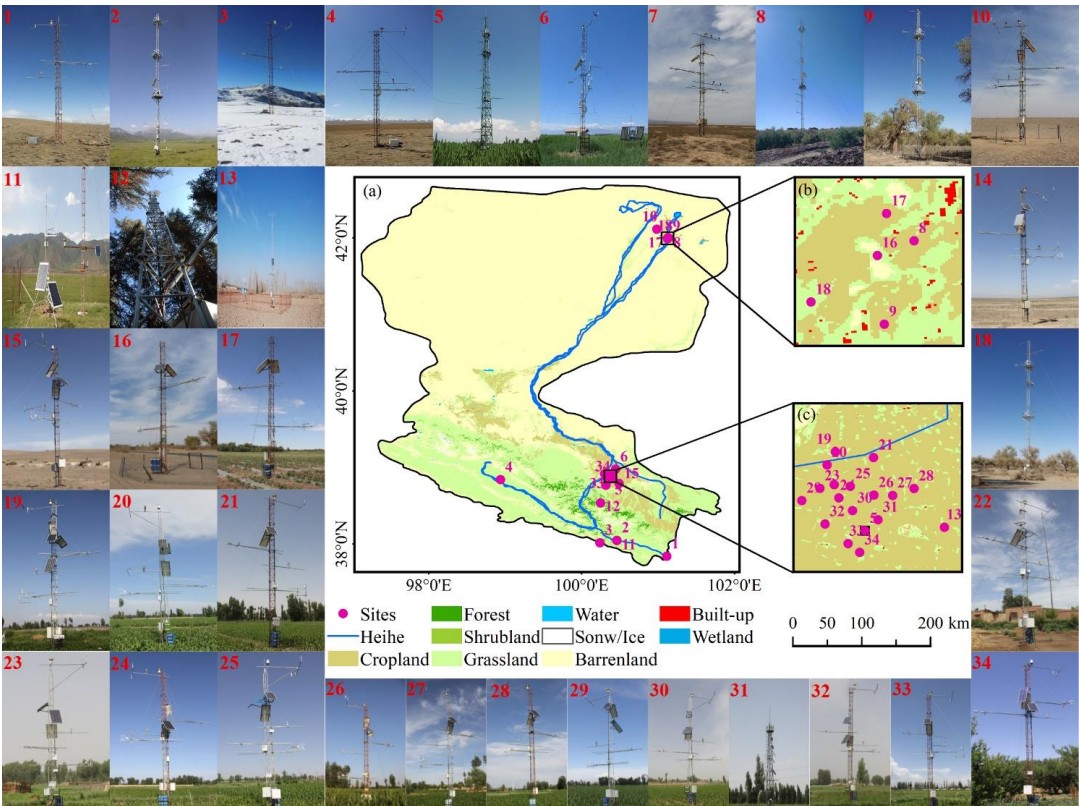

**Fig. 1 Distribution of eddy covariance (EC) observation sites in the Heihe River Basin (HRB).** (a) is EC sites distribution map in the Heihe River Basin with landcover as the background. (b) and (c) show the distribution of the EC sites in the matrix experiment area in middle reaches and core experiment area in the lower reaches. The photos from 1 to 34 illustrate the underlying landscapes of the EC sites.





**Table 1 Information of long-term and short-term observing sites in HRB.**

| ID | Name | Site_ID | Longitude (∘E) | Latitude (∘N) | Landcover | Dominant plant | Elevation (m) | Period | Stream |
|----|------|---------|----------------|---------------|-----------|----------------|---------------|--------|--------|
| 1 | Jingyangling site | JYL | 101.116 | 37.8384 | Alpine grassland | Kobresia pygmaea | 3750m | 2018- Now | Upper |
| 2 | Arou super site | ARS | 100.4643 | 38.0473 | Alpine grassland | Kobresia pygmaea | 3033m | 2013- Now | Upper |
| 3 | Dadongshu site | DDS | 100.2421 | 38.0142 | Alpine grassland | Kobresia pygmaea | 4148m | 2015- Now | Upper |
| 4 | Dashalong site | DSL | 98.9406 | 38.8399 | Alpine marshland | Kobresia pygmaea | 3739m | 2013- Now | Upper |
| 5 | Daman super site | DMS | 100.3722 | 38.8555 | Cropland | Seed corn | 1556m | 2012- Now | Middle |
| 6 | Zhangye wetland site | ZYW | 100.4464 | 38.9751 | Wetland | Reed | 1460m | 2012- Now | Middle |
| 7 | Huazhaizi site | HZZ | 100.3186 | 38.7652 | Desert | Salsola passerina | 1731m | 2012- Now | Middle |
| 8 | Sidaoqiao super site | SDQ | 101.1374 | 42.0012 | Woodland | Tamarisk | 873m | 2013- Now | Lower |
| 9 | Hunhelin site | HHL | 101.1335 | 41.9903 | Woodland | Populus euphratica and Tamarix | 874m | 2013- Now | Lower |
| 10 | Huangmo site | HMo | 100.9872 | 42.1135 | Desert | Reaumuria | 1054m | 2015- Now | Lower |
| 11 | Arou site | ARo | 100.4646 | 38.0443 | Alpine grassland | Kobresia pygmaea | 3033m | 2008-2011 | Upper |
| 12 | Guantan site | GTa | 100.2500 | 38.5333 | Subalpine forest | Picea crassifolia | 2835m | 2010-2011 | Upper |
| 13 | Yingke site | YKe | 100.4103 | 38.8571 | Cropland | Seed corn | 1519m | 2007-2011 | Middle |
| 14 | Bajitan site | BJT | 100.3042 | 38.9150 | Gobi | / | 1562m | 2012-2014 | Middle |
| 15 | Shenshawo site | SSW | 100.4933 | 38.7892 | Desert | / | 1594m | 2012-2015 | Middle |
| 16 | Luodi site | LDi | 101.1326 | 41.9993 | Bare land | / | 878m | 2013-2015 | Lower |
| 17 | Nongtian site | NTi | 101.1338 | 42.0048 | Cropland | Cucumis melo | 875m | 2013-2015 | Lower |
| 18 | Huyanglin site | HYL | 101.1239 | 41.9932 | Woodland | Populus euphratica forest | 876m | 2013-2015 | Lower |

**Table 2 Information of sites in the eddy covariance matrix experiment in the middle reaches of the HRB in 2012.**

| ID | Name | Site_ID | Longitude (∘E) | Latitude (∘N) | Landcover | Dominant plant | Elevation(m) | Period |
|----|------|---------|----------------|---------------|-----------|----------------|--------------|--------|
| 19 | EC Matrix 1 | M01 | 100.35813 | 38.89322 | Cropland | Vegetable | 1552.75m | 2012/6/4-9/17 |
| 20 | EC Matrix 2 | M02 | 100.35406 | 38.88695 | Cropland | Seed corn | 1559.09m | 2012/6/3-9/21 |
| 21 | EC Matrix 3 | M03 | 100.37634 | 38.89053 | Cropland | Seed corn | 1543.05m | 2012/6/3-9/18 |
| 22 | EC Matrix 4 | M04 | 100.35753 | 38.87752 | Built-up | / | 1561.87m | 2012/5/31-9/17 |
| 23 | EC Matrix 5 | M05 | 100.35068 | 38.87574 | Cropland | Seed corn | 1567.65m | 2012/6/3-9/18 |
| 24 | EC Matrix 6 | M06 | 100.35970 | 38.87116 | Cropland | Seed corn | 1562.97m | 2012/5/28-9/21 |
| 25 | EC Matrix 7 | M07 | 100.36521 | 38.87676 | Cropland | Seed corn | 1556.39m | 2012/5/29-9/18 |
| 26 | EC Matrix 8 | M08 | 100.37649 | 38.87254 | Cropland | Seed corn | 1550.06m | 2012/5/28-9/21 |
| 27 | EC Matrix 9 | M09 | 100.38546 | 38.87239 | Cropland | Seed corn | 1543.34m | 2012/6/4-9/17 |
| 28 | EC Matrix 10 | M10 | 100.39572 | 38.87567 | Cropland | Seed corn | 1534.73m | 2012/6/4-9/17 |
| 29 | EC Matrix 11 | M11 | 100.34197 | 38.86991 | Cropland | Seed corn | 1575.65m | 2012/5/29-9/18 |
| 30 | EC Matrix 12 | M12 | 100.36631 | 38.86515 | Cropland | Seed corn | 1559.25m | 2012/5/28-9/21 |
| 31 | EC Matrix 13 | M13 | 100.37852 | 38.8607 | Cropland | Seed corn | 1550.73m | 2012/5/27-9/20 |
| 32 | EC Matrix 14 | M14 | 100.35310 | 38.85867 | Cropland | Seed corn | 1570.23m | 2012/5/30-9/21 |



| 33 | EC Matrix 16 | M16 | 100.36411 | 38.84931 | Cropland | Seed corn | 1564.31m | 2012/6/6-9/17 |
| 34 | EC Matrix 17 | M17 | 100.36972 | 38.84510 | Cropland | Orchard | 1559.63m | 2012/5/31-9/17 |

## 3 Data post-processing

In this work, the data post-process included three steps: 1) to perform quality control on the auxiliary meteorological data and fill the gaps in the meteorological data by combining meteorological reanalysis data; 2) to conduct quality control of the NEE measurements and to fill the flux data gaps; 3) to partition the half-hourly NEE measurements to GPP and Reco. The flow diagram for the data post-processing is shown in Fig. 2.

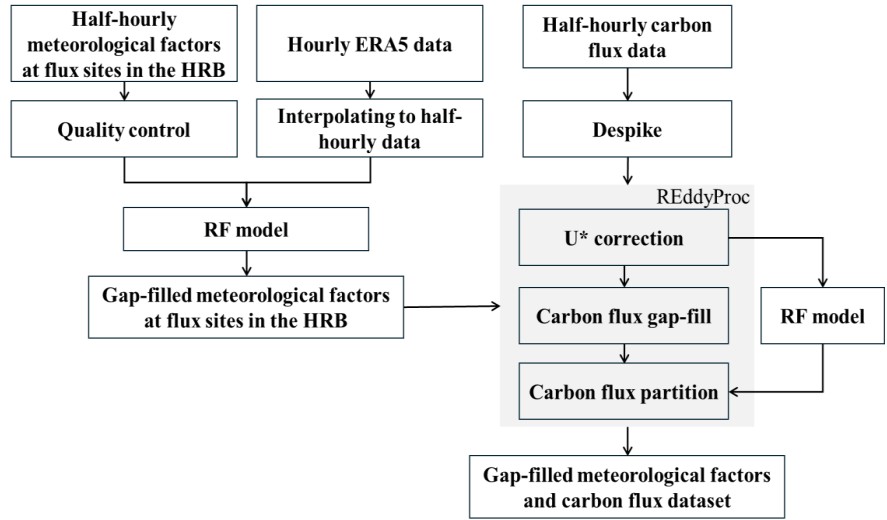

**Fig. 2 Diagram for the meteorological and carbon flux data post-processing in the HRB.**

### 3.1 Meteorological data post-processing

The half-hourly meteorological data underwent quality control to remove outliers, and only high-quality records were retained (Xu et al., 2020; Liu et al., 2018). In this work, we selected eight meteorological factors (Rg, Ta, Ts, RH, VPD, SWC, P, PA) measured at the flux sites in the HRB for post-processing, as these factors are highly related to carbon fluxes and are also available in the ERA5-Land (ECMWF Reanalysis v5) dataset. The ERA5-Land dataset is a global reanalysis dataset with spatial resolution of 0.1 degree and temporal resolution of 1 hour (Muñoz-Sabater et al., 2021). The corresponding factors were extracted from the ERA5-Land dataset according to the geographical coordinates of each site. To match the temporal resolution of the in-situ observed data, the extracted hourly ERA5-Land data was linearly interpolated to half-hourly data for Rg, Ta, Ts, RH, VPD, SWC, and PA. For precipitation (P), linear interpolation could result in overestimation of the yearly precipitation amount, and therefore the hourly precipitation was equally divided to two half-hours. After temporal matching between in-situ observations and extracted ERA5-Land data, a random forest (RF) model was trained for each factor. To test the accuracy of the RF model in meteorological gap-filling, 5-days of continuous artificial gaps were created in the meteorological factors, and these were then used to assess the performance of the gap-filling method. The RF model was able to accurately predict the missing meteorological observations for all variables except P using ERA5-land variables as input (Fig. 3).



**Fig. 3 The performance of RF models in the gap-filling of the meteorological data.** Rg: downward shortwave radiation; RH: relative humidity; SWC: soil water content; P: precipitation; PA: atmospheric pressure; VPD: vapor pressure deficit; Ta: air temperature; Ts: soil temperature. The suffix "_Obs" indicates the observed values. The suffix "_RF" indicates the random forest predicted values.



### 3.2 Carbon flux data post-processing

The original 10 Hz EC data were processed into 30-minute flux data by Xu and Liu (Xu et al., 2020; Liu et al., 2018).
Here, we further processed the 30-minute data to obtain continuous GPP, NEP, and Reco. First, outliers in the 30-minute NEE data were excluded based on a three-times standard deviation criterion (Rousseeuw and Croux, 1993), which is a widely used method in meteorological aberrant values detection.

Second, to further exclude poor-quality NEE data, u* filtering was applied using the REddyProc package (Wutzler et al., 2018), a post-processing tool for half-hourly EC measurements. During the night, stable stratification often occurs, leading to
underestimation of nighttime NEE. This issue was identified by examining the relationship between NEE and u*. Nighttime NEE values with u* lower than a threshold u* were filtered as invalid. A revised breakpoint detection method (Barr et al., 2013) in the REddyProc package was used to determine the threshold u*. After u* filtering, the gaps in the half-hourly data increased.

Third, it was necessary to fill these gaps to obtain continuous NEE data. In this study, both marginal distribution sampling
(MDS) (Reichstein et al., 2005) and RF were implemented to fill the gaps by combining gap-filled Rg, Ta, and VPD data with valid NEE data. The MDS method fills half-hourly NEE gaps using different schemes depending on the availability of meteorological data and is included in REddyProc package. For the RF method, a RF model is built using high-quality observed NEE and auxiliary meteorological factors (Rg, Ta, and VPD). This model is then used to fill the NEE gaps by inputting the gap-filled Rg, Ta, and VPD data. Both MDS and RF are effective in filling the gaps in NEE, with $R^2$ values of 0.77 for MDS
and 0.84 for RF between the filled and observed values (Fig. 4). While RF can fill all the gaps in NEE, MDS still leaves some long gaps unfilled (Fig. 5).

Fourth, the gap-filled NEE data were partitioned into GPP and Reco, two critical variables in carbon cycle studies. The NEE partitioning was also performed using the REddyProc package. During nighttime (Rg < 10 W/m$^2$), NEE equals Reco because there is no photosynthesis. The Lloyd-Taylor respiration function (Lloyd and Taylor, 1994) was fitted using nighttime
NEE and Ta. This fitted Lloyd-Taylor respiration function was then applied to estimate daytime Reco, with GPP calculated as the difference between Reco and NEE during the daytime (Wang et al., 2012).

To improve user convenience, the dataset variables were aggregated at multiple time intervals, including daily (_DD), weekly (_WW), monthly (_MM), and yearly (_YY). Variables such as GPP, Reco, NEE, and Precipitation were aggregated over longer intervals using the sum. In contrast, variables like Rg, Ta, Ts, SWC, RH, VPD, PA, LE, and H were aggregated
using the average.

Considering the close performance of the MDS and RF methods, and the fact that the RF method can fill all gaps in the data, the subsequent analysis of carbon flux in the HRB is based on the RF results.



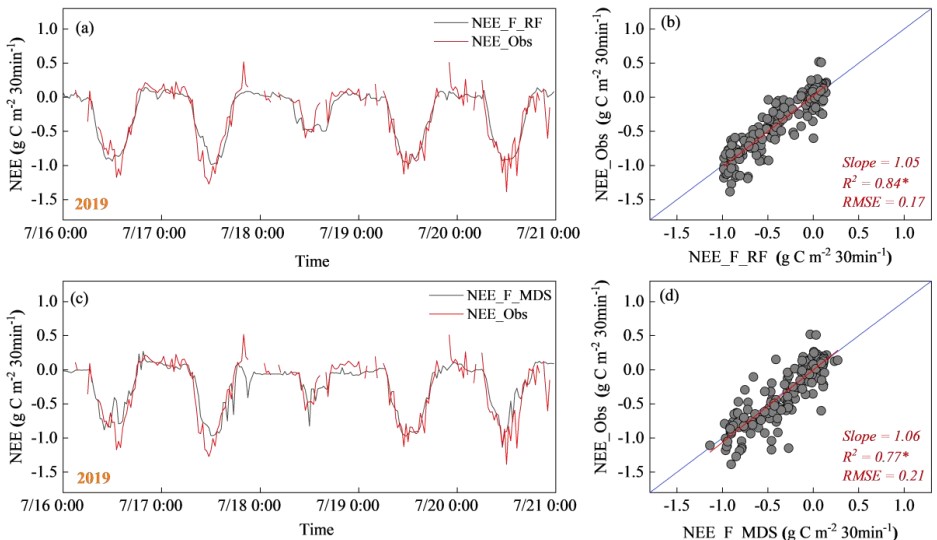

**Fig. 4 The performance of RF models in carbon flux gap-filling.** The suffix "_Obs" indicates the observed values.
The suffix "_RF" indicates the gap-filled values by random forest method. The suffix "_MDS" indicates the gap-filled values
by MDS method.

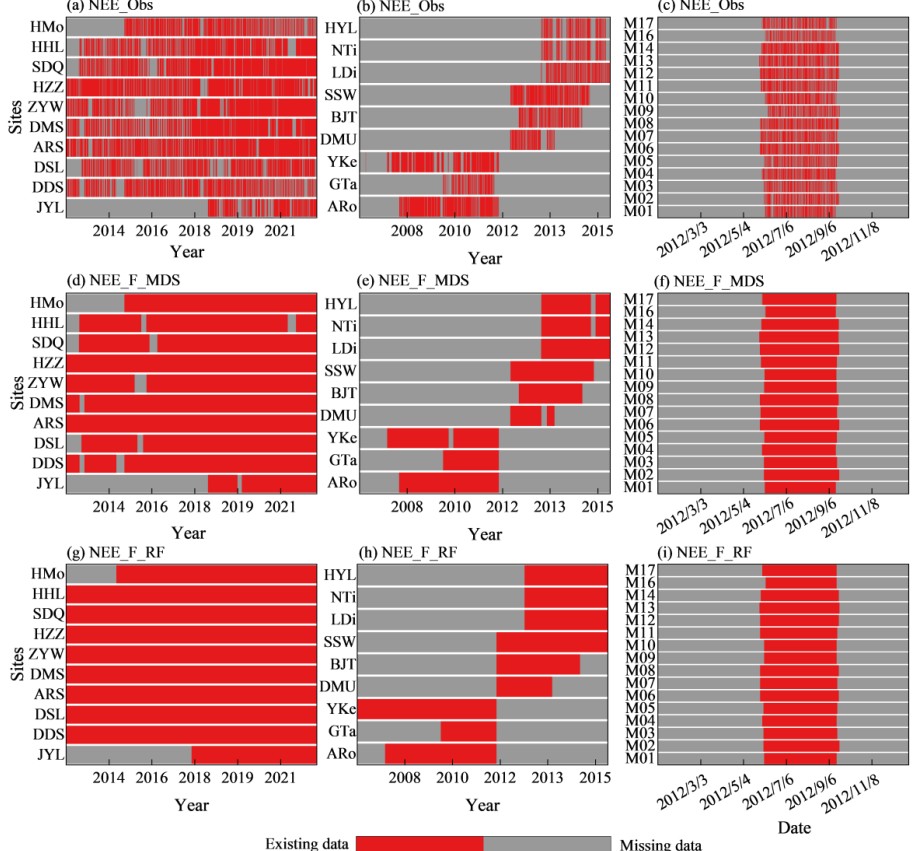

**Fig. 5 Data gaps in NEE before and after gap-filling.** The suffix "_Obs" indicates the observed values. The suffix "_RF"





indicates the gap-filled with random forest method. The suffix "_MDS" indicates the gap-filled with MDS method.

## 4 Dataset description

The post-processed dataset includes 34 sites with time spans ranging from a few months to ten years. For each site, the data comprises both original and gap-filled auxiliary meteorological factors (downward solar radiation: Rg, air temperature: Ta, soil temperature: Ts, relative humidity: RH, soil water content: SWC, precipitation: P, and atmospheric pressure: PA) and flux data (net ecosystem exchange: NEE, gross primary productivity: GPP, ecosystem respiration: Reco, latent heat flux: LE, and sensible heat flux: H). The data is provided at multiple temporal scales: half-hourly (_HH), daily (_DD), weekly (_WW), monthly (_MM), and yearly (_YY). The folder for each specific site is named according to the following convention: Site_ID + start year + end year + temporal scale suffix. The data is saved in the CSV format. Fields with the suffix "_ERA" indicate data extracted from ERA5-Land. Fields with the suffix "_F" represent gap-filled data. The explanation of the fields in the post-processed data is shown in Table 3.

**Table 3 Data fields in half-hourly dataset.**

| Variables | Description | Unit | |
|---|---|---|---|
| TIMESTAMP_START | The initial time of observation | - | |
| Year | Year | - | |
| DoY | The day of the year | - | |
| Hour | Hour of the day | - | |
| NEE | Net ecosystem exchange | g C/m²/30min | |
| LE | Latent heat flux | W m⁻² | |
| H | Sensible heat flux | W m⁻² | |
| Rg | Downward shortwave radiation | W m⁻² | In-situ observed data |
| Ta | Air temperature | degC | |
| Ts_1 | Surface soil temperature | degC | |
| RH | Relative humidity | % | |
| VPD | Saturated vapor pressure difference | hPa | |
| SWC_1 | Surface soil water content | % | |
| P | Precipitation | mm | |
| PA | Atmospheric pressure | hPa | |
| uStar | Friction wind speed | ms⁻¹ | |
| LE_ERA | Latent heat flux | W m⁻² | |
| H_ERA | Sensible heat flux | W m⁻² | |
| Rg_ERA | Downward shortwave radiation | W m⁻² | |
| Ta_ERA | Air temperature | degC | |
| Ts_1_ERA | Surface soil temperature | degC | Extracted ERA data |
| RH_ERA | Relative humidity | % | |
| VPD_ERA | Saturated vapor pressure difference | hPa | |
| SWC_1_ERA | Surface soil water content | % | |
| P_ERA | Precipitation | mm | |
| PA_ERA | Atmospheric pressure | hPa | |
| NEE_F_MDS | Gap-filled NEE with MDS method | g C/m²/30min | |
| NEE_F_RF | Gap-filled NEE with RF method | g C/m²/30min | |
| NEE_F_fqc | Quality flag for Gap-filled NEE | - | |
| H_F_MDS | Gap-filled sensible heat flux with MDS method | W m⁻² | |
| H_F_RF | Gap-filled sensible heat flux with RF method | W m⁻² | |
| H_F_fqc | Quality flag for H | - | Gap-filled data |
| LE_F_MDS | Gap-filled latent heat flux with MDS method | W m⁻² | |
| LE_F_RF | Gap-filled latent heat flux with RF method | W m⁻² | |
| LE_F_fqc | Quality flag for LE | - | |
| Rg_F_RF | Downward shortwave radiation | W m⁻² | |
| Rg_F_fqc | Air temperature | - | |

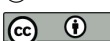

| | | | |
|---|---|---|---|
| Ta_F_RF | Air temperature | degC | |
| Ta_F_fqc | Quality flag for Ta | - | |
| Ts_1_F_RF | Surface soil temperature | degC | |
| Ts_1_F_fqc | Quality flag for Ts | - | |
| RH_F_RF | Relative humidity | % | |
| RH_F_fqc | Quality flag for RH | - | |
| VPD_F_RF | Saturated vapor pressure difference | hPa | |
| VPD_F_fqc | Quality flag for VPD | - | |
| SWC_1_F_RF | Surface soil water content | % | |
| SWC_1_F_fqc | Quality flag for SWC | - | |
| P_F_RF | Precipitation | mm | |
| P_F_fqc | Quality flag for P | - | |
| PA_F_RF | Atmospheric pressure | hPa | |
| PA_F_fqc | Quality flag for PA | - | |
| GPP_F_RF | Gross primary production partitioned from NEE_F_RF | g C/m²/30min | |
| Reco_F_RF | Ecosystem respiration partitioned from NEE_F_RF | g C/m²/30min | Partitioned |
| GPP_F_MDS | Gross primary production partitioned from NEE_F_MDS | g C/m²/30min | Carbon flux |
| Reco_F_MDS | Ecosystem respiration partitioned from NEE_F_MDS | g C/m²/30min | |

*fqc in HH data: 1 = measured; 0 = gap-filled; fqc in DD, WW, MM and YY data: indicating percentage of missed data (0-1).

## 5 Results

### 5.1 Diurnal variations of carbon fluxes for various ecosystem types in the HRB

To explore the temporal dynamics of the carbon fluxes for diverse ecosystem types in the HRB, the carbon fluxes were averaged for different ecosystems, including subalpine forest, alpine grassland, cropland, wetland, riparian woodland and gobi/desert. The averaged diurnal cycle curves and statistic metrics of these ecosystems in growing season (May to September) are shown in Fig. 6 and Table 4. Note that negative NEE values indicate net carbon uptake, while positive NEE indicate net carbon release. Over the course of the diurnal cycle, NEE, GPP and Reco varied greatly for subalpine forest, alpine grassland,

cropland and wetland but slightly for gobi/desert. The half-hourly NEE reached minimum (i.e., the largest net carbon uptake) at 11:30pm for subalpine forest, alpine grassland and riparian woodland, 12:30pm for cropland, 13:00pm for wetland and 11:30pm for gobi/desert. The GPP varied greatly in subalpine forest, alpine grassland, cropland, wetland and riparian woodland, and kept constant close to zero in gobi/desert. The half-hourly GPP reached maximum at 11:30pm for subalpine forest and alpine grassland, at 12:30 for riparian woodland, cropland and gobi/desert, at 13:00pm for wetland. The half-hourly Reco

reached maximum at 16:30pm for subalpine forest and gobi/desert, at 17:00 for riparian woodland, at 15:30pm for alpine grassland, at 16:00pm for cropland and wetland (Table 4 and Fig. 6).



Earth System
Open Access  Science  Discussions
Data

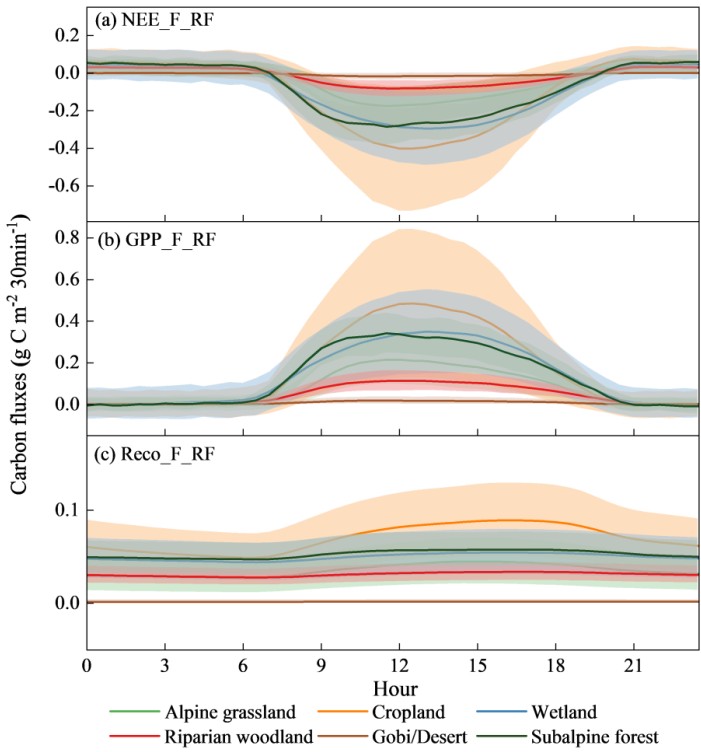

**Fig. 6 Diurnal variation in the growing season of carbon fluxes of different ecosystems in the HRB.**

**Table 4 Statistical metrics of carbon flux diurnal curves for ecosystems in the HRB.** (NEE$_{Min}$ is minimum NEE in
diurnal course and NEE$_{Min}$ Time is corresponding time. GPP$_{Max}$ is maximum GPP in diurnal course and GPP$_{Max}$ Time is the
corresponding time. Reco$_{Max}$. is maximum Reco in diurnal course and Reco$_{Max}$ Time is the corresponding time. Unit for
NEE$_{Min}$, GPP$_{Max}$ and Reco$_{Max}$ is gC m$^{-2}$ 30min$^{-1}$)

|  | Subalpine forest | Riparian woodland | Alpine grassland | Cropland | Wetland | Gobi/Desert |
|---|---|---|---|---|---|---|
| NEE$_{Min}$ | -0.29 | -0.081 | -0.17 | -0.40 | -0.30 | -0.017 |
| NEE$_{Min}$ Time | 11:30 | 11:30 | 11:30 | 12:30 | 13:00 | 11:30 |
| GPP$_{Max}$ | 0.34 | 0.113 | 0.22 | 0.49 | 0.35 | 0.0187 |
| GPP$_{Max}$ Time | 11:30 | 12:30 | 11:30 | 12:30 | 13:00 | 12:30 |
| Reco$_{Max}$ | 0.06 | 0.034 | 0.045 | 0.089 | 0.055 | 0.002 |
| Reco$_{Max}$ Time | 16:30 | 17:00 | 15:30 | 16:00 | 16:00 | 16:30 |

**5.2 Seasonal variation of carbon fluxes for ecosystems in the HRB**

The eighteen sites (Table 1) with more than one year of data were selected to explore the seasonal dynamics of carbon
fluxes for different ecosystems in the HRB. These sites were grouped into six ecosystem types. The seasonal dynamics of
carbon fluxes for these ecosystems are shown in Fig. 7.

Seasonal NEE varied significantly throughout the year for subalpine forest, alpine grassland, wetland, cropland, and
riparian woodland but remained close to 0 gC m$^{-2}$ day$^{-1}$ year-round for gobi/desert. During the non-growing season, NEE was
close to zero for all ecosystems except forest. In the transition period from non-growing to growing season, NEE slightly
increased and became positive. During the growing season, NEE was notably less than zero for all ecosystems except
gobi/desert, indicating that these ecosystems except gobi/desert exhibited net carbon uptake. The minimum NEE for all
ecosystems occurred in July, with values of -5.62 gC m$^{-2}$ day$^{-1}$ for subalpine forest, -1.48 gC m$^{-2}$ day$^{-1}$ for riparian woodland,
-4.08 gC m$^{-2}$ day$^{-1}$ for alpine grassland, -12.40 gC m$^{-2}$ day$^{-1}$ for cropland, -8.24 gC m$^{-2}$ day$^{-1}$ for wetland, and -0.49 gC m$^{-2}$ day$^{-}$



[1] for gobi/desert (Table 5).

Seasonal GPP also varied significantly throughout the year. During the non-growing season, GPP was very close to zero for all ecosystems except for subalpine forest. During the growing season, GPP was obviously higher than zero for all ecosystems except gobi/desert. Seasonal GPP reached its maximum value in July, with values of 8.11 gC m$^{-2}$ day$^{-1}$ for subalpine forest, 3.08 gC m$^{-2}$ day$^{-1}$ for riparian woodland, 6.62 gC m$^{-2}$ day$^{-1}$ for alpine grassland, 16.84 gC m$^{-2}$ day$^{-1}$ for cropland, 10.98 gC m$^{-2}$ day$^{-1}$ for wetland, and 0.75 gC m$^{-2}$ day$^{-1}$ for gobi/desert (Table 5).

Seasonal Reco followed a temporal pattern similar with seasonal GPP. Reco also reached its maximum in July, with values of 3.93 gC m$^{-2}$ day$^{-1}$ for subalpine forest, 1.98 gC m$^{-2}$ day$^{-1}$ for riparian woodland, 2.96 gC m$^{-2}$ day$^{-1}$ for alpine grassland, 5.64 gC m$^{-2}$ day$^{-1}$ for cropland, 3.41 gC m$^{-2}$ day$^{-1}$ for wetland, and 0.17 gC m$^{-2}$ day$^{-1}$ for gobi/desert (Table 5).

**Table 5 Statistical metrics of carbon flux seasonal curves for ecosystems in the HRB.** (NEE$_{Mean}$ and NEE$_{Min}$ are daily average and minimum NEE in over the year. GPP$_{Mean}$ and GPP$_{Max}$ are daily average and minimum GPP in over the year.

Reco$_{Mean}$ and Reco$_{Max}$ are daily average and minimum GPP in over the year. Unit for NEE$_{Mean}$, NEE$_{Min}$, GPP$_{Mean}$, GPP$_{Max}$, Reco$_{Mean}$ and Reco$_{Max}$ is gC m$^{-2}$ day$^{-1}$.)

| | Subalpine forest | Riparian woodland | Alpine grassland | Cropland | Wetland | Gobi/Desert |
|---|---|---|---|---|---|---|
| NEE$_{Mean}$ | -2.10 | -0.34 | -0.84 | -1.75 | -1.83 | -0.25 |
| NEE$_{Min}$ | -5.62 | -1.48 | -4.08 | -12.40 | -8.24 | -0.49 |
| GPP$_{Mean}$ | 3.52 | 1.18 | 1.67 | 3.50 | 3.08 | 0.30 |
| GPP$_{Max}$ | 8.11 | 3.08 | 6.62 | 16.84 | 10.98 | 0.63 |
| Reco$_{Mean}$ | 1.42 | 0.84 | 0.83 | 1.75 | 1.25 | 0.05 |
| Reco$_{Max}$ | 3.93 | 1.98 | 2.96 | 5.64 | 3.41 | 0.17 |

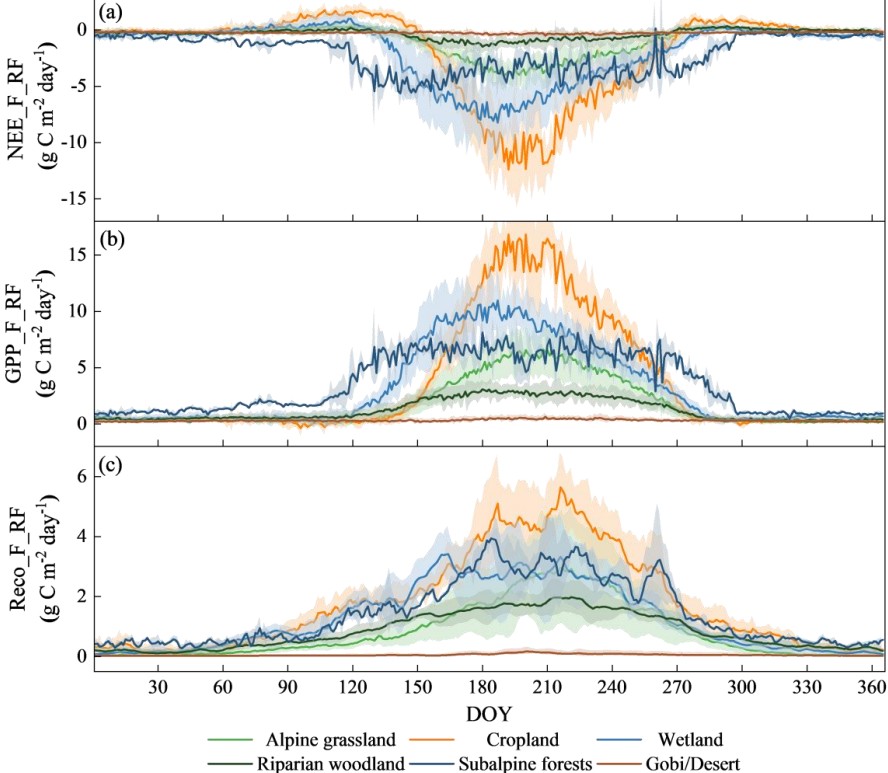

**Fig. 7 Seasonal dynamics of carbon fluxes for different ecosystem types in the HRB.**





### 5.3 Inter-annual variations of carbon fluxes for various ecosystem types in the HRB

To explore the inter-annual variations of carbon fluxes in different ecosystem types of the HRB, nine sites with more than seven years of data were selected. These sites were grouped into five ecosystem types. The yearly dynamics of GPP, Reco, and NEE is shown in Fig. 8, and the statistical metrics of GPP, Reco, and NEE are provided in Table 6.

The multi-year average NEE was -123.43 gC m$^{-2}$ year$^{-1}$ in riparian woodland, -307.84 gC m$^{-2}$ year$^{-1}$ in alpine grassland, -638.77 gC m$^{-2}$ year$^{-1}$ in cropland, -679.62 gC m$^{-2}$ year$^{-1}$ in wetland and -92.04 gC m$^{-2}$ year$^{-1}$ in gobi/desert. Yearly NEE was the highest in cropland and the lowest in gobi/desert. Annual NEE of wetlands significantly increased during the last decade, while other ecosystems exhibited relatively stable NEE with slight inter-annual variations (Fig. 8a). The multi-year average GPP was 431.47 gC m$^{-2}$ year$^{-1}$ for riparian woodland, 609.22 gC m$^{-2}$ year$^{-1}$ for alpine grassland, 1269.19 gC m$^{-2}$ year$^{-1}$ for cropland, 1127.88 gC m$^{-2}$ year$^{-1}$ for wetlands, and 108.22 gC m$^{-2}$ year$^{-1}$ for gobi/desert. Over the last decade, the annual GPP of wetland and riparian woodland slightly increased, while the GPP of other ecosystems remained relatively stable (Fig. 8b). The multi-year average Reco was 308.03 gC m$^{-2}$ year$^{-1}$ for riparian woodland, 301.38 gC m$^{-2}$ year$^{-1}$ for alpine grassland, 630.42 gC m$^{-2}$ year$^{-1}$ for cropland, 448.26 gC m$^{-2}$ year$^{-1}$ for wetlands, and 16.18 gC m$^{-2}$ year$^{-1}$ for gobi/desert. The Reco in the HRB slightly increased for cropland and wetlands but remained relatively stable for other ecosystem types over the last decade (Fig. 8c).

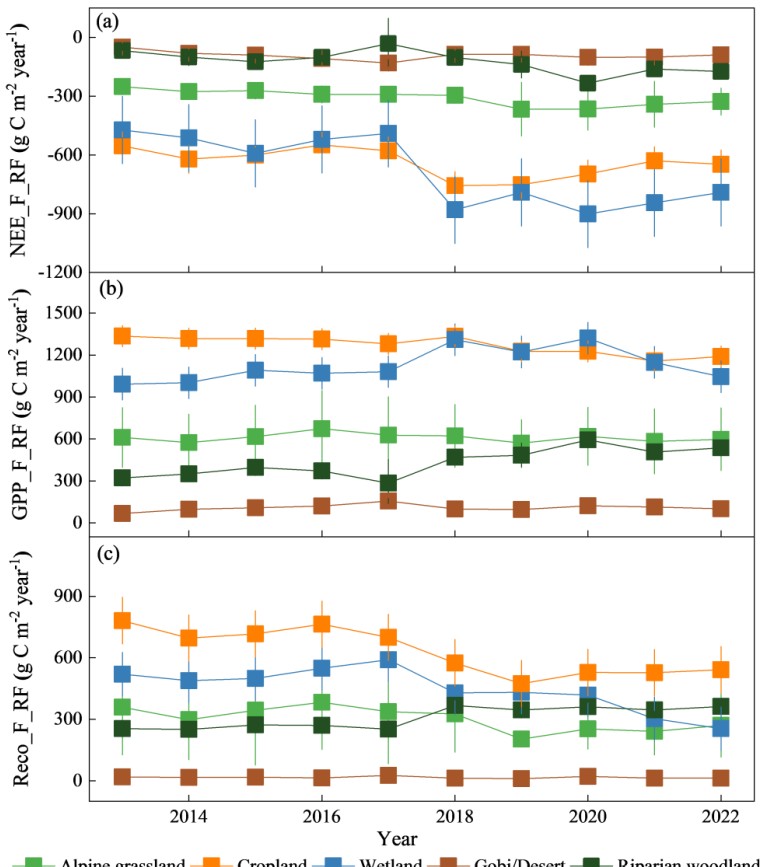

**Fig. 8 Yearly NEE, Reco and GPP for the main ecosystems in the HRB.**

### 5.4 Spatial variations of carbon fluxes in the HRB



Earth System
Science
Data

To examine the spatial patterns of carbon fluxes, the annual NEE, GPP and Reco of the eighteen sites with at least 1 year of data were compared. Annual NEE, GPP and Reco of the eighteen sites in the HRB are shown in Fig. 9. In the upper reaches of the river basin, annual NEE and GPP was the highest at the GTa site and the lowest at the DDS site. In the middle reaches, NEE and GPP was the highest at the ZYW, DMS and YKe sites, which are in the artificial oasis. In the lower reaches, NEE and GPP were higher at HHL, HYL, SDQ and NTi than at HMo and LDi. In the upper reaches, the NEE, GPP and Reco generally increased with elevation. In the middle and lower reaches, NEE, GPP and Reco of the sites inside the artificial/natural oasis were obviously higher than those of the sites outside the artificial/natural oasis.

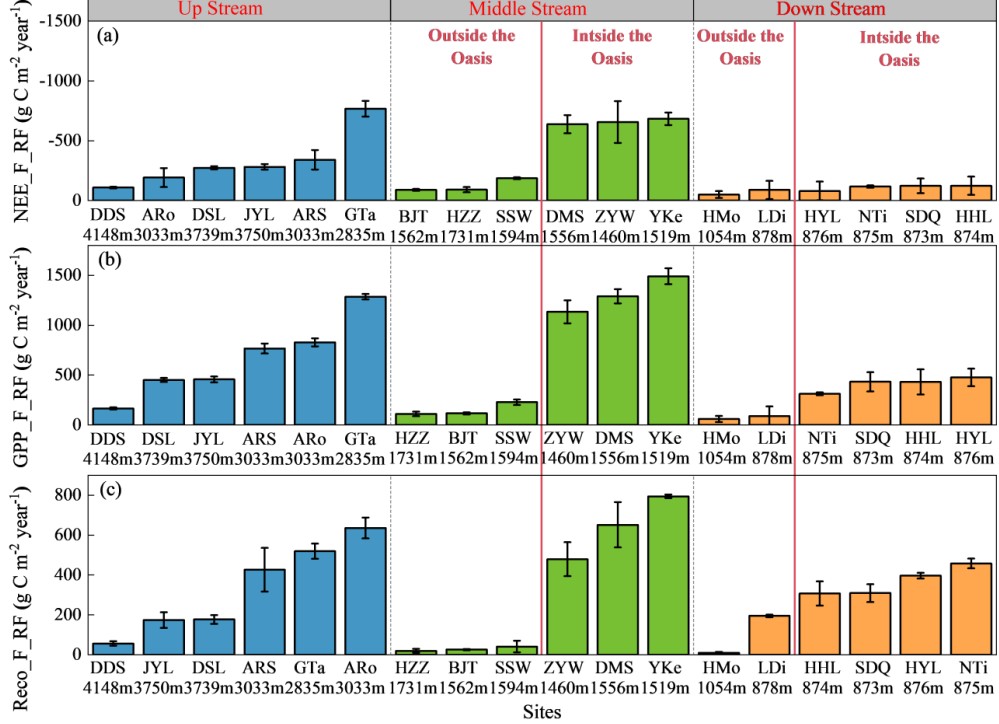

**Fig. 9 Yearly carbon fluxes of the 18 sites with more than 1 year of data for the upper, middle, and lower reaches of the HRB.** The number under each site ID indicates the elevation of the site.

To explore the carbon flux change along the environmental gradients, we sorted the carbon flux sites based on air temperature, precipitation, soil water content, and downward shortwave radiation. We then explored the carbon flux variation in relation to these four factors. The yearly average air temperature increases from the upper reaches to the middle and lower reaches of the HRB. However, the GPP, Reco, and NEE did not show a similar gradient pattern with air temperature. The GPP, Reco, and NEE are significantly higher at DMS, YKe, GTa, and ZYW than at other sites, and the temperature at these sites is at an intermediate level among all the sites (Fig. 10a). In the upper reaches of the HRB, the NEE, GPP and Reco of the sites decreased as annual average temperature decreased, while in the middle and lower reaches of the HRB, NEE, GPP and Reco did not change with the temperature gradient. NEE, GPP and Reco generally follow the same spatial pattern, with higher carbon fluxes at sites with higher soil water content. The cropland and wetland with irrigation in the middle reaches have the highest GPP, Reco and NEE (Fig. 10b). Precipitation decreases from about 500 mm in the upper reaches to about 50 mm in the lower reaches of the HRB. The GPP, Reco, and NEE of these sites did not strictly increase or decrease with the spatial precipitation gradient (Fig. 10c). GPP, Reco, and NEE did not change with the gradient of Rg among the sites in the HRB (Fig. 10d).



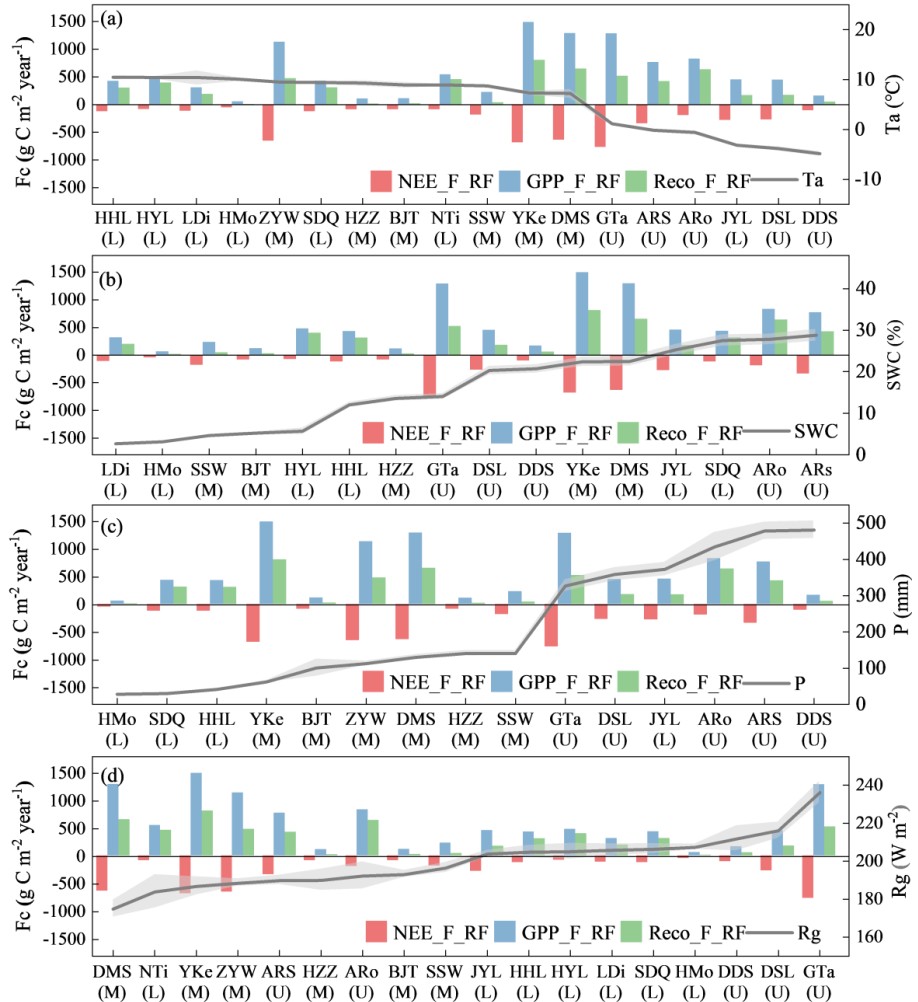

**Fig. 10 Carbon flux spatial variation with the gradients of meteorological factors.** Letters in the bracket stand for subregions in the HRB, U for upper reaches, M for middle reaches and L for lower reaches of the HRB.

**6 Discussion**

**6.1 Carbon flux pattern and its drivers in the HRB**

EC-based carbon flux data is perhaps the most effective data source to quantify the carbon sequestration capacity of ecosystems at the ecosystem scale. As the HRB is a typical inland river basin, the temporal and spatial patterns of carbon fluxes in this region provide insight into the carbon dynamics of inland river basins in the Northwest China and Central Asia

more broadly.

The diurnal pattern of carbon flux shows that GPP and NEE peak around midday when downward shortwave radiation is at its highest, while Reco reaches its peak later in the afternoon when temperatures are at their maximum. This indicates that during the growing season, the diurnal NEE curve is primarily driven by GPP variations. The GPP diurnal pattern is largely influenced by downward shortwave radiation, whereas the Reco pattern is controlled by temperature (Kato et al., 2004). The

half-hourly GPP and NEE of subalpine forest, riparian woodland, alpine grassland, and gobi/desert peak slightly earlier than those of cropland and wetlands. This difference is likely due to the fact that cropland and wetlands are not limited by water or



heat, whereas the other ecosystems experience stress from either heat or water (Lin et al., 2019).

The seasonal patterns of NEE, GPP, and Reco vary significantly among different ecosystems. In gobi/desert, NEE, GPP, and Reco remain close to zero throughout the year due to sparse vegetation coverage and low soil organic matter content. For alpine grassland, NEE, GPP, and Reco begin to increase later in spring and decrease earlier in autumn compared to subalpine forest, wetland, and riparian woodland. This can be attributed to the lower temperatures and higher elevations of alpine grassland (Wang et al., 2022). In cropland, NEE, GPP, and Reco also show a late start in spring and an early decline in autumn, mainly driven by agricultural management practices (Guo et al., 2021).

The upper reaches of the HRB are humid and cold, and the alpine grassland are weak carbon sinks with annual NEE range from -108 gC m$^{-2}$ year$^{-1}$ to -341.56 gC m$^{-2}$ year$^{-1}$. This is consistent with the previous studies on alpine grassland at the Haibei site (Zhao et al., 2005) and the Dangxiong site (Shi et al., 2006). The subalpine forest (picea crassifolia) in the upper reaches is strong carbon sink, with NEE of -767.01 gC m$^{-2}$ year$^{-1}$, which has been rarely reported. The carbon flux in the upstream region of the HRB is mainly stressed by low temperature (Sun et al., 2019).

The middle reaches of the HRB are dry and hot, with significant differences in carbon fluxes between sites inside and outside the artificial oasis. Sites within the artificial oasis have strong carbon uptake capacity, with NEE exceeding -600 gC m$^{-2}$ year$^{-1}$ due to irrigation. In contrast, sites outside the artificial oasis have very weak carbon uptake capacity, with NEE less than -100 gC m$^{-2}$ year$^{-1}$. Inside the artificial oasis, high temperatures and high soil water content promote vegetation growth, while outside the artificial oasis, high temperatures and low soil water content inhibit vegetation growth. Due to intensive irrigation, the carbon fluxes of sites inside the artificial oasis are decoupled from precipitation in this region (Wang et al., 2019).

The lower reaches are even drier and hotter than the middle reaches in the HRB. The NEE of the sites in this region ranges from -49.72 gC m$^{-2}$ year$^{-1}$ to -123.85 gC m$^{-2}$ year$^{-1}$. The natural oasis in the lower reaches consist of riparian ecosystems distributed along the main river channels. Vegetation in the natural oasis survives by relying on lateral water supply from the river channel and shallow groundwater. Vegetation in the natural oasis faces slightly water stress compared to vegetation in the artificial oasis in the middle reaches. The vegetation outside the natural oasis in the lower reaches faces more severe water stress than vegetation outside the artificial oasis in the middle reaches due to lower precipitation and soil water availability in the downstream region.

**6.2 Possible sources of uncertainty in the carbon data in the HRB**

Data post-processing can introduce uncertainties into carbon flux measurements. Due to instrument malfunctions and maintenance, data gaps are inevitable, yet continuous carbon flux data is essential for assessing an ecosystem's carbon uptake capacity. In post-processing, three key steps—u* correction, gap-filling, and carbon flux partitioning—can result in uncertainties. The u* correction can filter a large proportion of nighttime carbon flux data, with different methods yielding different u* thresholds and varying proportions of filtered data. This correction can impact data availability for building lookup tables or training models in the gap-filling process.

The gap-filling process uses mathematical methods aided by meteorological data to fill in missing data, which can introduce significant uncertainties. The MDS and random forest methods are the two primary techniques currently used for data gap-filling (Zhu et al., 2022), and both are evaluated in this work for the HRB. The performance of MDS and random forest methods is very close, and both can effectively fill gaps in half-hourly NEE data. While previous studies have reported that MDS may systematically overestimate carbon emissions and underestimate $CO_2$ sequestration (Vekuri et al., 2023), we did not observe this phenomenon in the HRB. However, MDS cannot effectively fill gaps longer than two weeks, whereas the RF method can fill all gaps if the corresponding auxiliary meteorological data are available.

The NEE partitioning method can also introduce uncertainties into GPP and Reco data (Tramontana et al., 2020). Although the night-time-based method is recommended as a standard by FLUXNET, it has some limitations. It only considers



temperature in the respiration estimation, neglecting other environmental factors that could influence respiration. Additionally, it does not account for variations in respiration between nighttime and daytime under different light conditions (Oikawa et al., 2017; Raj et al., 2016; Chen et al., 2024). Since direct measurement of GPP and Reco are not available, assessing uncertainties in the NEE partitioning step remains challenging.

In addition to uncertainties introduced by data processing, harsh weather, complex terrain, and instrument maintenance can contribute to uncertainties in carbon flux observations in the HRB. In the upstream regions, extremely cold temperatures
during the non-growing season can occasionally lead to frost formation on the gas analyzer, affecting the $CO_2$ concentration signals. In the midstream and downstream areas, sandstorms can disrupt the optical path of the gas analyzer. Furthermore, finding large, flat, ideal locations for EC instrumentation can be challenging for certain ecosystems. For example, subalpine forest in the upstream area are primarily located on shaded hill slopes, suggesting that carbon fluxes at the GTa site may require additional processing to account for terrain effects. Additionally, instrument degradation and updates can introduce
uncertainties or inconsistencies into the original observation data.

**7 Data use guidelines**

Data are fully public but should be appropriately referenced by citing this paper and the database (see Sect. 8). We suggest that researchers planning to use this dataset as a core dataset for their analysis contact and collaborate with the first or corresponding authors of this paper.

**8 Data availability**

The post-processed carbon flux and auxiliary data in the HRB is available at: https://doi.org/10.11888/Terre.tpdc.301321 (Wang et al., 2024).

**9 Conclusions**

Over the past decade, a comprehensive carbon flux network has been established in the Heihe River Basin (HRB) in
Northwest China. In this study, carbon flux and auxiliary meteorological data from the network were post-processed to create an analysis-ready dataset. This dataset encompasses 34 sites across six dominant ecosystems in the HRB: alpine grassland, subalpine forest, cropland, wetland, riparian woodland, and gobi/desert. Eighteen of these sites have continuous multi-year observations, while 16 sites were observed only during the 2012 growing season, totaling 1,513 site-months. Based on this dataset, the following temporal and spatial characteristics of carbon exchange in the HRB were identified: 1) In the diurnal
variation curve, GPP, NEE, and Reco peak later for ecosystems in the artificial oasis (cropland and wetland) compared to those outside the artificial oasis (grassland, forest, woodland, and gobi/desert). 2) Seasonal NEE, GPP, and Reco peak in early July for grassland, forest, woodland, and cropland, while remain close to zero throughout the year for gobi/desert. 3) In the last decade, NEE of wetlands significantly increased, while NEE for other ecosystems slightly fluctuated inter-annually. 4) NEE, GPP, and Reco are significantly higher for sites inside the artificial/natural oasis compared to those outside. This post-
processed carbon flux dataset has many applications, e.g., exploring carbon exchange characteristic of alpine and arid ecosystem, ecosystem responses to climate extremes, cross-site synthesis at regional to global scales, regional and global upscaling studies, interpreting and calibrating remote sensing products, evaluating and calibrating carbon cycle models.

**Author contributions**

X.F. Wang, T. Che, J.F. Xiao, X. Li and S.M. Liu designed the research. X.F. Wang and T.H. Wang processed and
analyzed the data. J.L. Tan, Y. Zhang, Z.G. Ren, Z.W. Xu, L.Y. Geng and H.B. Wang collected the data, maintained carbon flux network in HRB and processed the original data. X.F. Wang, T. Che, T.H. Wang and J.F. Xiao wrote the paper.



**Competing interests**

The contact author has declared that none of the authors has any competing interests.

**Acknowledgements**

We thank all the scientists, engineers, and students who contributed to the establishment and maintenance of the carbon flux network in the Heihe River Basin.

**Financial support**

This work has been supported by the Gansu Provincial Science and Technology Program (22ZD6FA005), the Foundation for Distinguished Young Scholars of Gansu Province (Grant No. 22JR5RA046), the Funds of National Natural

Science Foundation of China (Grant No. 42371386 and U22A202690), and the Youth Innovation Promotion Association CAS to X.W. (No. 2020422). J.F. Xiao was supported by University of New Hampshire via bridge support.

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
