# Peer review of "A post-processed carbon flux dataset for 34 eddy covariance flux sites across the Heihe River Basin, China"

_Earth System Science Data, 2024_

## Community Comment (CC1)

The study *"Post-processed carbon flux dataset for 34 eddy covariance flux sites across the Heihe River Basin, China"* presents a post-processed carbon flux dataset derived from 34 eddy covariance (EC) sites in the Heihe River Basin, China. This dataset is expected to be invaluable for studying carbon cycling in arid regions and Asia, where substantial data gaps exist due to limited FLUXNET observations and data sharing. I strongly support the publication of this study, pending minor revisions as outlined below:

**Line 26**: Please clarify the term "multi-year observation" to help readers quickly determine the applicability of the dataset to their research.

**Abstract**: Adding a data access portal at the end of the abstract is recommended to facilitate wider use of this dataset.

**Line 191**: Check the time format—before 12:00 should be labeled "AM." Alternatively, consider removing "AM" and "PM" entirely since a 24-hour format is already used.

**Figure 7**: It is suggested to translate the X-axis labels from DOY (Day of Year) to specific month names for better readability.

**Line 243**: The statement "The Reco in the HRB slightly increased for cropland and wetlands but remained relatively stable for other ecosystem types over the last decade" appears inconsistent with Figure 8c, which shows that Reco in cropland and wetlands is declining. Please clarify.

**Lines 62–63**: Regarding the claim, "The GPP, Reco, and NEE are significantly higher at DMS, YKe, GTa, and ZYW than at others," how do the authors explain the behavior of Reco at the Aro station?

**Lines 323–325**: The text states, "While previous studies have reported that MDS may systematically overestimate carbon emissions and underestimate CO2 sequestration (Vekuri et al., 2023), we did not observe this phenomenon in the HRB." Please elaborate on how this result was derived.

**Line 331**: The statement "Since direct measurement of GPP and Reco are not available, assessing uncertainties in the NEE partitioning step remains challenging" should be revised as there are indeed methods for directly measuring Reco, such as the chamber technique.

---

## Author Comment (AC1)

**Response to reviewers' comments (Paper # essd-2024-370)**

We have fully considered the reviewer's comments during the revision and have improved the manuscript accordingly. We summarize our responses point by point below in blue. The revised places are highlighted in yellow background in the revised manuscript.

Reviewer comments:

The study *"Post-processed carbon flux dataset for 34 eddy covariance flux sites across the Heihe River Basin, China"* presents a post-processed carbon flux dataset derived from 34 eddy covariance (EC) sites in the Heihe River Basin, China. This dataset is expected to be invaluable for studying carbon cycling in arid regions and Asia, where substantial data gaps exist due to limited FLUXNET observations and data sharing. I strongly support the publication of this study, pending minor revisions as outlined below:

Response: We thank the reviewer for the positive evaluation of our manuscript.

**Line 26**: Please clarify the term "multi-year observation" to help readers quickly determine the applicability of the dataset to their research.

Response: We have revised the "multi-year observation" to " multi-year observation during 2008-2022".

**Abstract**: Adding a data access portal at the end of the abstract is recommended to facilitate wider use of this dataset.

Response: As per your recommendation, we have included the data access portal at the end of the abstract.

**Line 191**: Check the time format—before 12:00 should be labeled "AM." Alternatively, consider removing "AM" and "PM" entirely since a 24-hour format is already used.

Response: Thank you for the suggestions. We have updated the time format to the 24-hour format and revised it accordingly throughout the manuscript.

**Figure 7**: It is suggested to translate the X-axis labels from DOY (Day of Year) to specific month names for better readability.

Response: Thank you for the suggestions. We have revised the X-axis labels of Fig.7.

**Line 243**: The statement "The Reco in the HRB slightly increased for cropland and wetlands but remained relatively stable for other ecosystem types over the last decade" appears inconsistent with Figure 8c, which shows that Reco in cropland and wetlands is declining. Please clarify.

Response: Thank you to the reviewer for kindly pointing this out. We made a typographical error here: the Reco slightly decreased for croplands and wetlands, as shown in Figure 8c.

**Lines 262–263**: Regarding the claim, "The GPP, Reco, and NEE are significantly higher at DMS,

YKe, GTa, and ZYW than at others," how do the authors explain the behavior of Reco at the Arou station?

Response: Thank you for your comments. The ARo site is unique, exhibiting high GPP and Reco but low NEE. We guess that it may be attributed to the high soil organic content at this location. And need further study to confirm. For the YKe, DMS, GTa, and ZYW sites, which represent cropland, forest, and wetland ecosystems, they demonstrate high GPP, Reco, and NEE. We have also revised Figure 9 to use a consistent value range for the three carbon fluxes, which makes it easier to read.

**Lines 323–325**: The text states, "While previous studies have reported that MDS may systematically overestimate carbon emissions and underestimate $CO_2$ sequestration (Vekuri et al., 2023), we did not observe this phenomenon in the HRB." Please elaborate on how this result was derived.

Response: In a previous study (Vekuri et al., 2023), it was reported that the MDS method could introduce significant errors in carbon balance estimates. In this study, we also evaluated the performance of the MDS method by introducing artificial gaps into the NEE observations. Our results did not reveal any systematic errors in the gap-filled data generated using the MDS method, as illustrated in Fig. 4.

**Line 331**: The statement "Since direct measurement of GPP and Reco are not available, assessing uncertainties in the NEE partitioning step remains challenging" should be revised as there are indeed methods for directly measuring Reco, such as the chamber technique.

Response: Thank you for your comments. We agree that the original statement is not rigorous. For ecosystems with short vegetation, Reco can be measured using chamber techniques. However, for ecosystems with tall vegetation, such as forests, measuring Reco is challenging. Accordingly, we have revised the statement to:

"Since direct measurements of GPP and Reco are difficult, especially in ecosystems with tall vegetation, assessing uncertainties in the NEE partitioning process remains challenging."

---

## Author Comment (AC2)

**Response to reviewers' comments (Paper # essd-2024-370)**

We have fully considered the reviewer's comments during the revision and have improved the manuscript accordingly. We summarize our responses point by point below in blue. The revised places are highlighted in yellow background in the revised manuscript.

Reviewer comments:

In this manuscript, Wang et al compiled and post-processed half-hourly flux data from these 34 EC flux sites in the Heihe River Basin to create a continuous, homogenized time series dataset. This work filled the gaps in carbon flux data and auxiliary meteorological data of the Heihe carbon flux network, and generated a carbon flux dataset. Then the half-hourly NEE measurements were partitioned into GPP and Reco. The diurnal, seasonal, and inter-annual variabilities of carbon flux across diverse ecosystems in the Heihe River Basin were explored based on the gap-filled and partitioned dataset. This post-processed carbon flux dataset with a total of 34 EC sites in the Heihe River Basin is very valuable and important.

Response: We sincerely appreciate the reviewer's positive feedback on our dataset.

Overall, I find the paper compelling and fit for publication after minor revision. I only have a few comments as below:

ERA5-Land dataset is a reanalysis dataset. It would be great if some evaluation for variables from ERA5-land vs observations can be added.

Response: Thank you for your comments. We have evaluated the ERA5-Land variables against observational data and found that ERA5-Land variables exhibit systematic errors (as shown in Fig. 1 of this response letter). These systematic errors are inconsistent; for instance, ERA5-Land Ta and VPD closely match observations in the morning but are significantly lower than observations in the afternoon.

Given these errors and the spatial scale mismatch between ERA5-Land data and observations, we developed an RF model using ERA5 data and local observations. The RF model accurately predicts local observations based on ERA5 data, as demonstrated in Fig. 3 of the manuscript. This model is then utilized to effectively fill gaps in the observed meteorological data.

Considering the length of the article, we did not include the figures related to the evaluation of ERA5 in the manuscript.

[Figure]

Fig.1 Comparison ERA5-land variables vs observation

If possible, comparing the MDS with RF results will be interesting.

Response: Actually, we compared the performance of the MDS method with the RF method. Specifically, in section 3.2, we introduced artificial gaps into the NEE data, and both MDS and RF effectively filled these gaps, with the RF method slightly outperforming MDS. In some cases, MDS struggled to fill longer gaps, while RF was able to fill these gaps more effectively. This comparison has also been discussed in section 6.2.

The RF model was used for meteorological and carbon-flux data post-processing. The readers will be curious to know the construction and parameter selection of the RF model.

Response: Thank you for the comments. We have added the details for RF model construction for meteorological and carbon-flux gap-filling in section 3.1 and 3.2, respectively.

批注 [XW1]: 增加相应的描述

In section 3.1:

*In the RF model, the tree number (n_estimators) was set 800, the random_state was set to 30, test sample size was set to 0.3, other parameters kept at their default values as provided in the sklearn package.*

In section 3.2:

*The RF model settings are identical to those used for meteorological data gap-filling.*

Please give some explanation why only Rg, Ta, and VPD are selected as input of the RF model to predict NEE.

Response: Since the MDS method also uses Rg, Ta, and VPD to fill the gaps, we built the RF gap-filling model using the same three variables. Meanwhile, these three variables are easily accessible. We have added an explanation for selecting Rg, Ta, and VPD in section 3.2, which is highlighted with a yellow background.

*Since the MDS method uses Rg, Ta, and VPD to fill the gaps, we built the RF gap-filling model using the same three variables. Additionally, Rg, Ta, and VPD are the main factors that control ecosystem carbon exchange.*

---

## Author Comment (AC3)

Response to reviewers' comments (Paper # essd-2024-370)

We have fully considered the reviewer's comments during the revision and have improved the manuscript accordingly. We summarize our responses point by point below in blue. The revised places are highlighted in yellow background in the revised manuscript.

Reviewer comments:

In this manuscript, Wang et al compiled and post-processed half-hourly flux data from these 34 EC flux sites in the Heihe River Basin to create a continuous, homogenized time series dataset. This work filled the gaps in carbon flux data and auxiliary meteorological data of the Heihe carbon flux network, and generated a carbon flux dataset. Then the half-hourly NEE measurements were partitioned into GPP and Reco. The diurnal, seasonal, and inter-annual variabilities of carbon flux across diverse ecosystems in the Heihe River Basin were explored based on the gap-filled and partitioned dataset. This post-processed carbon flux dataset with a total of 34 EC sites in the Heihe River Basin is very valuable and important.

Response: We sincerely appreciate the reviewer's positive feedback on our dataset.

Overall, I find the paper compelling and fit for publication after minor revision. I only have a few comments as below:

ERA5-Land dataset is a reanalysis dataset. It would be great if some evaluation for variables from ERA5-land vs observations can be added.

Response: Thank you for your comments. We have evaluated the ERA5-Land variables against observational data and found that ERA5-Land variables exhibit systematic errors (as shown in Fig. 1 of this response letter). These systematic errors are inconsistent; for instance, ERA5-Land Ta and VPD closely match observations in the morning but are significantly lower than observations in the afternoon.

Given these errors and the spatial scale mismatch between ERA5-Land data and observations, we developed an RF model using ERA5 data and local observations. The RF model accurately predicts local observations based on ERA5 data, as demonstrated in Fig. 3 of the manuscript. This model is then utilized to effectively fill gaps in the observed meteorological data.

Considering the length of the article, we did not include the figures related to the evaluation of ERA5 in the manuscript.

[Figure]

Fig.1 Comparison ERA5-land variables vs observation

If possible, comparing the MDS with RF results will be interesting.

Response: Actually, we compared the performance of the MDS method with the RF method. Specifically, in section 3.2, we introduced artificial gaps into the NEE data, and both MDS and RF effectively filled these gaps, with the RF method slightly outperforming MDS. In some cases, MDS struggled to fill longer gaps, while RF was able to fill these gaps more effectively. This comparison has also been discussed in section 6.2.

The RF model was used for meteorological and carbon-flux data post-processing. The readers will be curious to know the construction and parameter selection of the RF model.

Response: Thank you for the comments. We have added the details for RF model construction for meteorological and carbon-flux gap-filling in section 3.1 and 3.2, respectively.

In section 3.1:

*In the RF model, the tree number (n_estimators) was set 800, the random_state was set to 30, test sample size was set to 0.3, other parameters kept at their default values as provided in the sklearn package.*

In section 3.2:

*The RF model settings are identical to those used for meteorological data gap-filling.*

Please give some explanation why only Rg, Ta, and VPD are selected as input of the RF model to predict NEE.

Response: Since the MDS method also uses Rg, Ta, and VPD to fill the gaps, we built the RF gap-filling model using the same three variables. Meanwhile, these three variables are easily accessible. We have added an explanation for selecting Rg, Ta, and VPD in section 3.2, which is highlighted with a yellow background.

*Since the MDS method uses Rg, Ta, and VPD to fill the gaps, we built the RF gap-filling model using the same three variables. Additionally, Rg, Ta, and VPD are the main factors that control ecosystem carbon exchange.*